# Unsupervised neural machine translation with generative language models only

## Abstract

We show how to derive state-of-the-art unsupervised neural machine translation systems from generatively pre-trained language models. Our method consists of three steps: *few-shot amplification*, *distillation*, and *backtranslation*. We first use the zero-shot translation ability of large pretrained language models to generate translations for a small set of unlabeled sentences. We then amplify these zero-shot translations by using them as few-shot demonstrations for sampling a larger synthetic dataset. This dataset is then distilled by discarding the few-shot demonstrations and then fine-tuning. During backtranslation, we repeatedly generate translations for a set of inputs and then fine-tune a single language model on both directions of the translation task at once, ensuring cycle-consistency by swapping the roles of gold monotext and generated translations when fine-tuning. By using our method to leverage GPT-3's zero-shot translation capability, we achieve a new state-of-the-art in unsupervised translation on the WMT14 English-French benchmark, attaining a BLEU score of 42.1.

## 1 Introduction

Recent work on generative pre-training has shown that with sufficient data and scale (Kaplan et al., 2020; Henighan et al., 2020; Radford et al., 2019), large language models (LMs) acquire remarkable in-context metalearning abilities (Brown et al., 2020). One of the most striking ways this capability manifests is via *few-shot learning*, where the model picks up patterns from multiple training examples placed in context. While few-shot prompting is flexible and enables strong performance on a diverse suite of NLP tasks to be coaxed out of generatively pre-trained LMs, its benefits are most pronounced with larger models, with commensurate training, inference, compute, and data costs. The desire to reduce these costs motivates our present work, which allows us to continue finetuning our models, obtaining more performance from smaller models and pushing our larger models even further, without resorting to few-shot prompting at test time or any additional supervision at train time.

We target the domain of unsupervised neural machine translation (NMT), which typically involves *bootstrapping* a weak translation model before amplifying its translation ability via *backtranslation*. Recent work in unsupervised NMT has been dominated by large encoder-decoder architectures where the bootstrap is implemented by denoising/autoencoding tasks (*e.g.*, multilingual Cloze (Devlin et al., 2019; Conneau & Lample, 2019), masked-span prediction (Raffel et al., 2020; Xue et al., 2021), reconstruction from corrupted inputs (Wang et al., 2019; Liu et al., 2020)) intended to produce strong encoders and aligned multilingual representations for decoding. In our present work, we show that generative language modeling alone can implement the entire unsupervised NMT pipeline, and derive state-of-the-art unsupervised NMT systems using only generatively pre-trained language models. We implement the bootstrap by first sampling a small number of zero-shot translations from GPT-3. These are then used as few-shot prompts to sample a larger dataset of synthetic translations. The few-shot prompts are then discarded and the generated samples are *distilled* by fine-tuning the model on these synthetic data in the zero-shot format. This produces a language model aligned to our translation format and amenable to large-scale backtranslation. By using our method to leverage GPT-3's zero-shot translation capability, we achieve a new state-of-the-art in unsupervised translation on the WMT14 English-French benchmark, attaining a BLEU score of 42.1.

## 2 BACKGROUND AND RELATED WORK

The modern approach to unsupervised neural machine translation typically involves encoder-decoder architectures jointly trained via denoising autoencoding / reconstruction tasks (Vincent et al., 2008; Conneau & Lample, 2019; Liu et al., 2020; Ma et al., 2020; Raffel et al., 2020; Xue et al., 2021; Wang et al., 2019; Liu et al., 2020; Song et al., 2019) and backtranslation (Sennrich et al., 2016; Edunov et al., 2018; Cotterell & Kreutzer, 2018). This approach to unsupervised NMT is codified by Artetxe et al. (2018) and Lample et al. (2018), although various ideas can be traced back further: unsupervised machine translation was framed as a deciphering task by Ravi & Knight (2011) and backtranslation was first introduced for machine translation as a method for data augmentation using target-side monolingual data by Sennrich et al. (2016). Denoising autoencoding with a bilingual encoder can be viewed as a kind of latent bilingual lexicon induction, necessary for producing sufficiently aligned embeddings to kick-start backtranslation; such techniques have been extensively studied in the context of machine translation (Artetxe et al., 2017; Klementiev et al., 2012; Vulic & Moens, 2015; Hu et al., 2017; Goyal et al., 2016; Shen et al., 2017).

At the same time, recent work on large-scale generative pre-training (Kaplan et al., 2020; Henighan et al., 2020; Radford et al., 2019) has demonstrated that with sufficient data and model scale, strong performance on a diverse suite of NLP tasks can be coaxed from transformer language models using few-shot prompts. Our present work unifies these two lines of research by using generative language modeling to simplify unsupervised NMT even further: we show how with sufficient scale, pre-training, and clever prompting, a single generative language model can implement the entire unsupervised neural machine translation pipeline, avoiding optimizations such as denoising autoencoding, auxiliary / adversarial losses in latent space, or ad-hoc bilingual dictionaries.

Our reliance on large-scale generative pre-trainingis similar to prior work in unsupervised NMT which uses large-scale language modeling tasks on internet data as part of the bootstrap (Conneau & Lample, 2019; Conneau et al., 2020; Liu et al., 2020). The role of few-shot prompting and distillation in our method is related to recent work on unsupervised data augmentation using language models (Anaby-Tavor et al., 2020; Schick et al., 2021; Kumar et al., 2020; Papanikolaou & Pierleoni, 2020; Schick & Schütze, 2021; Yang et al., 2020) and is also in the same spirit as recent work on *self-training* and *noisy-student training* (Mi et al., 2021; Vu et al., 2021; Xie et al., 2020). The few-shot distillation component of our method is similar to contemporaneous work by Wang et al. (2021b) which uses few-shot prompting for unsupervised data augmentation, though they focus only on inference for text classification rather than generation for sequence-to-sequence tasks like machine translation and they do not study the phenomena of self-amplification nor few-shot data efficiency (Section 6) as we do.

## 3 BACKTRANSLATION VIA LANGUAGE MODELING

---

**Algorithm 1** Iterated backtranslation using a single generative language model

---

**Input:** Source monotext $\mathcal{M}_S$; target monotext $\mathcal{M}_T$; number of iterations $I$; number of samples per iteration $J$; monotext formatter $f(\cdot)$; bitext formatter $g(\cdot, \cdot)$; parameters $\boldsymbol{\theta}$ of language model $p_{\boldsymbol{\theta}}(\cdot)$ trained to complete outputs of $f$ to outputs of $g$.

**Output:** Final model parameters $\boldsymbol{\theta}$.

1: **for** $i = 1$ **to** $I$ **do**
2:      $\mathcal{B}_{back} \leftarrow \emptyset$
3:      **for** $j = 1$ **to** $J$ **do**
4:          $\mathbf{y} \sim \mathcal{M}_S \cup \mathcal{M}_T$
5:          $\tilde{\mathbf{x}} \sim p_{\boldsymbol{\theta}}(\cdot \mid f(\mathbf{y}))$
6:          $\mathcal{B}_{back} \leftarrow \mathcal{B}_{back} \cup \{\langle \tilde{\mathbf{x}}, \mathbf{y} \rangle\}$
7:      estimate $\boldsymbol{\theta}$ by maximizing $\log p_{\boldsymbol{\theta}}$ of $g(\tilde{\mathbf{x}}, \mathbf{y})$ for $\langle \tilde{\mathbf{x}}, \mathbf{y} \rangle \in \mathcal{B}_{back}$

---

Backtranslation was first introduced in the context of machine translation as a method for data augmentation using target-side monolingual data (Bojar & Tamchyna, 2011; Sennrich et al., 2016; Poncelas et al., 2018), by sampling synthetic source-to-target data from another target-to-source translation model. In our present work, we cast machine translation as a language modeling task and

jointly train and sample from a single language model for both source-to-target and target-to-source translation.

Given bitext ⟨`seq1`, `seq2`⟩ in languages $L_1$ and $L_2$, we format the translation task as follows:



`[L1] <seq1> [[TRANSLATE]] [L2] <seq2>`



At test-time, the LM is prompted with `[L1] <seq> [[TRANSLATE]] [L2]` and we parse a candidate translation `<sampledSeq>` from the sampled completion. Backtranslation is implemented by reversing the roles of `seq` and `sampledSeq` and finetuning on the bitext ⟨`sampledSeq, seq`⟩.

We remark that in contrast to the interpretation of backtranslation as a wake-sleep algorithm (Cotterell & Kreutzer, 2018), where the forwards and backwards translators are trained alternately, we use a single language model for both forwards and backwards translation and train on both directions jointly at every iteration.

There are various ways to train a model using backtranslation, *e.g.*, completely online (interleaving minibatch gradient updates and sampling) versus offline (backtranslating the entire training dataset at each epoch; potentially re-training the model from scratch after sampling new backtranslations). In practice, we find that data scaling of a model's optimal test loss and BLEU score quickly saturates on backtranslations from previous versions of the model, and opt for a semi-online setup where we synchronously sample a relatively small number of $L_1$-$L_2$ and $L_2$-$L_1$ pairs before resuming training for a single epoch on the newly sampled data. We refer to this as a single *iteration* of backtranslation.

Formally, Algorithm 1 describes our implementation of backtranslation using a single generative language model $p_{\boldsymbol{\theta}}(\cdot)$. We assume that $p_{\boldsymbol{\theta}}(\cdot)$ has already been trained to complete formatted monotext (`[L1] <seq1> [[TRANSLATE]] [L2]`) to formatted bitext (`[L1] <seq1> [[TRANSLATE]] [L2] <seq2>`).

## 4 THE BOOTSTRAP: GENERATIVE PRE-TRAINING, FEW-SHOT AMPLIFICATION, AND DISTILLATION

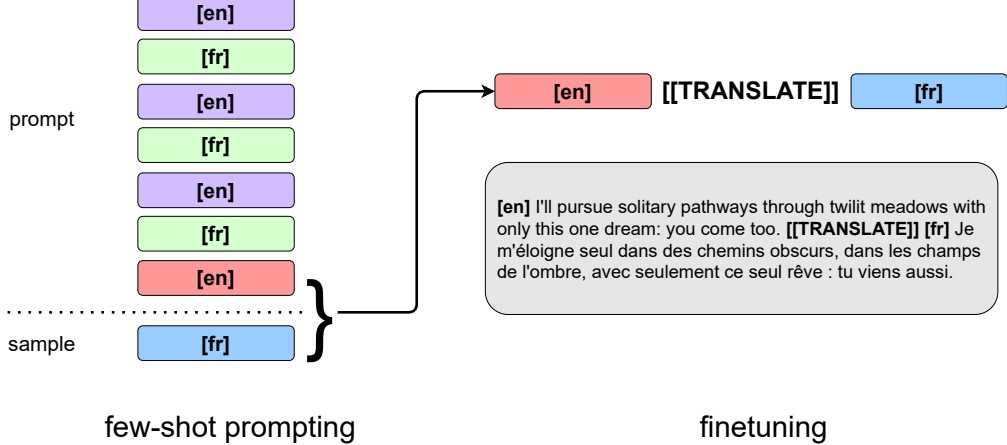

Figure 1: Illustration of our bootstrap procedure, which we call *few-shot distillation*. We use few-shot prompts sampled from GPT-3 to generate an initial dataset of synthetic translations from a generatively pretrained language model (left). The few-shot examples are then discarded and the synthetic bitext reformatted for finetuning on the autoregressive language modeling objective (right).

The modern approach to unsupervised NMT is parametrized by a choice of initialization or *bootstrap*. The bootstrap has typically relied on some form of unsupervised cross-lingual representation learning, *e.g.*, bilingual dictionaries initialized from unsupervised cross-lingual word embeddings (Lample et al., 2018; Artetxe et al., 2018) or multilingual masked language modeling followed by denoising autoencoding with a shared encoder and decoder (Conneau & Lample, 2019).

In Section 3, we formulated iterative backtranslation in terms of language modeling, assuming a language model which has already been trained to follow a particular instruction format for translation. To complete our procedure, we must supply such a language model. Unlike previous work on unsupervised NMT, we use language models from the GPT-3 family (Brown et al., 2020) which have been *generatively pre-trained* on a large corpus of Internet data. A key observation from the body of work around GPT-3 is that generative pre-training at scale induces strong in-context metalearning abilities, two special cases of which are (1) instruction following and (2) few-shot prompting: a sufficiently trained large language model benefits from both detailed natural language descriptions of tasks and, when given in-context examples, can achieve strong performance on a diverse suite of tasks (*e.g.*, question-answering, natural language inference, translation.) We implement the bootstrap by exploiting both of these abilities, by using natural language instruction to produce zero-shot translations and few-shot prompting during amplification.

## 4.1 FEW-SHOT AMPLIFICATION AND DISTILLATION

It thus remains to adapt our generatively pre-trained models' few-shot translation ability to the zero-shot format specified in Section 3. We do this in a two-stage process. We first sample a small number of zero-shot translations from GPT-3. Given bitext ⟨srcSeq, tgtSeq⟩ in srcLang and tgtLang, and a stop-sequence <sep>, we use the following format for zero-shot prompting:

```
<sep> Given the following passage in <srcLang>: <sep> <srcSeq> <sep>
a good <tgtLang> translation is: <sep> <tgtSeq> <sep>.
```

At test-time, we sample a completion until the stop-sequence <sep> is detected; throughout we set <sep> to be \n---\n.

We *amplify* these zero-shot translations by using them as few-shot prompts to sample a much larger synthetic dataset from a smaller model. We then *distill* this dataset by discarding the few-shot prompts and fine-tuning on formatted bitext, producing a language model aligned with our task format and amenable to backtranslation. In detail, we implement the bootstrap as follows:

1. Generatively pre-train a language model $p_{\boldsymbol{\theta}}(\cdot)$ on a large corpus of Internet data.

2. Sample a pool of $N_S$ synthetic target-side translations and $N_S$ target-side translations zero-shot from another language model $q(\cdot)$ for few-shot prompting. Using $k$ few-shot examples randomly drawn from $N_S$ (resp. $N_T$), sample $C_S$ synthetic target-side translations (resp. $C_T$ synthetic source-side translations) from $p_{\boldsymbol{\theta}}(\cdot)$, using the monolingual source-side corpus $\mathcal{M}_S$ (resp. target-side corpus $\mathcal{M}_T$).

3. Discard the few-shot prompts, reformat the (gold prompt, sampled translation) data as specified in Section 3, and finetune the language model $p_{\boldsymbol{\theta}}(\cdot)$ on these data.

4. Reverse all data and continue finetuning the language model $p_{\boldsymbol{\theta}}(\cdot)$ on the backtranslations (sampled translation, gold prompt).

**Why amplify and distill?** While few-shot prompting is flexible and enables strong performance on a diverse suite of NLP tasks to be coaxed out of generatively pre-trained LMs, its benefits are most pronounced with larger models, with commensurate training, inference, compute, and data costs. It is also unclear how to iteratively finetune a language model in a way that preserves its few-shot ability while remaining aligned with a zero-format like in Section 3. Few-shot amplification allows us to generate data for the bootstrap in an unsupervised fashion, possibly avoiding the overhead of few-shot sampling from GPT-3 itself by few-shot prompting a smaller model $p_{\boldsymbol{\theta}}(\cdot)$, while distillation enables iterative backtranslation.

## 5 RESULTS

**Experimental setup** For our experiments, we focus on the well-studied WMT14 English-French benchmark. In the notation of Algorithm 1, we obtain source and target monotext $\mathcal{M}_S$ and $\mathcal{M}_T$ by splitting the WMT14 English-French training set in half, each with approximately twenty million examples, and use only the English text from one half and only French text from the other to avoid implicit sentence-level alignment between source and target monotext. At each iteration of

backtranslation, we sample one million translations in either direction, *i.e,*. $J = 2e6$, and train for one epoch on the newly sampled data. For all of our results, unless otherwise specified, we run 40 iterations of backtranslation after the bootstrap and report BLEU using the final model checkpoint.

To implement the bootstrap, we additionally set aside 2048 training examples, and sample $N_S = 1024$ English-French (resp. $N_T = 1024$ French-English) translations zero-shot from GPT-3 to use as few-shot prompts. During few-shot amplification, we sample four million initial target- and source-side translations respectively using few-shot prompts, *i.e.*, $C_S = C_T = 4e6$ in the notation of Section 4.1, drawing monolingual prompts from as $\mathcal{M}_S$ and $\mathcal{M}_T$ defined above. We finetune for two epochs in the forwards direction (distillation) and for another two epochs in the backwards direction (initial backtranslation). For few-shot prompting, we use $k = 3$ in-context examples. In Section 6.3.1 we will see that we can minimize the number of few-shot examples to $N_S = N_T = 3$ with little effect on evaluation BLEU score after iterative backtranslation.

We use the same training setup and BPE tokenizer as GPT-3. During finetuning, we use a constant learning rate of $0.05 \cdot \ell$, where $\ell$ is the pre-training learning rate, a weight decay of $0.1$, and residual dropout $0.1$. When sampling during the bootstrap or during backtranslation, we default to using temperature $\tau = 0.3$. We ablate other values of $\tau$ in Section 6.1.

We report BLEU score on the official WMT14 English-French test set with greedy (argmax) sampling and sacreBLEU[1] (Post, 2018). In Table 3 we give a comparison to previous work on unsupervised NMT using `multi-bleu.perl` and the XLM (Conneau & Lample, 2019) tokenizer.

### 5.1 FEW-SHOT SELF-DISTILLATION AND BACKTRANSLATION

|  |  | small | medium | large | xl |
|---|---|---|---|---|---|
| few-shot ($\tau = 0.0$) | en-fr | 1.15 | 7.71 | 13.07 | 14.28 |
|  | fr-en | 5.04 | 16.87 | 20.25 | 23.0 |
| few-shot ($\tau = 0.3$) | en-fr | 1.02 | 7.36 | 11.89 | 13.58 |
|  | fr-en | 4.46 | 16.13 | 20.7 | 22.07 |
| few-shot ($\tau = 1.0$) | en-fr | 0.25 | 2.12 | 2.68 | 3.38 |
|  | fr-en | 1.22 | 5.45 | 6.14 | 9.32 |
| distillation | en-fr | 0.61 | 9.51 | 17.68 | 22.19 |
|  | fr-en | 4.31 | 23.67 | 29.38 | 31.12 |
| initial backtranslation | en-fr | 7.94 | 29.84 | 33.59 | 34.71 |
|  | fr-en | 1.5 | 23.12 | 28.58 | 30.52 |
| after backtranslation | en-fr | 30.48 | 36.53 | 37.59 | 39.12 |
|  | fr-en | 27.24 | 32.15 | 34.79 | 35.43 |

Table 1: English-French (top) and French-English (bottom) test BLEU throughout the few-shot self-distillation bootstrap across multiple model scales.

We first report results using *self-distillation*, *i.e.*, where during the bootstrap (Section 4) we sample from a single model which is then trained to imitate and then backtranslate its own few-shot prompted generations; for these experiments, the few-shot demonstrations themselves are generated zero-shot by GPT-3. This is then followed by the iterative backtranslation procedure described in Section 3. We apply this methodology to the `small`, `medium`, `large`, and `xl` models from the GPT-3 family (Brown et al., 2020), with 125M, 350M, 760M, and 1.3B parameters respectively. Table 1 displays test BLEU throughout our procedure for all model sizes. We see that translation out of English benefits significantly from the backtranslation part of the bootstrap alone. We also see that our models are much stronger at the translation task compared to few-shot prompting after only self-distillation. Finally, all models benefit significantly from iterative backtranslation, with English-French BLEU always converging to a slightly higher value than the reverse direction.

---

[1]SacreBLEU signature: `BLEU+case.mixed+numrefs.1+smooth.exp+tok.intl+version.1.2.20`.

## 5.2 DISTILLING SELF-AMPLIFIED GPT-3 INTO SMALLER MODELS

| | | small | medium | large | xl |
|---|---|---|---|---|---|
| distillation | en-fr | 34.13 | 36.03 | 37.21 | 37.08 |
| | fr-en | 32.34 | 34.96 | 36.12 | 36.34 |
| initial backtranslation | en-fr | 34.71 | 36.31 | 38.89 | 39.05 |
| | fr-en | 30.95 | 33.73 | 35.16 | 36.51 |
| after backtranslation | en-fr | 35.62 | 37.79 | 38.91 | 39.79 |
| | fr-en | 31.28 | 34.08 | 35.57 | 35.97 |
| after backtranslation (+CC100) | en-fr | 39.02 | 41.31 | 41.97 | **42.08** |
| | fr-en | 33.43 | 35.69 | 36.85 | **37.09** |

Table 2: English-French (top) and French-English (bottom) test BLEU throughout the bootstrap and after iterative backtranslation, this time using generations from self-amplified GPT-3 for the bootstrap. We observe the best performance by mixing in monotext from the English and French components of the CC100 dataset (Wenzek et al., 2020; Conneau et al., 2020) during backtranslation.

Although we do not apply our full methodology to the 175B parameter GPT-3 model due to compute constraints, we observe that for few-shot distillation, instead of training a model on few-shot samples from itself, we can just as well distill on few-shot samples from a much larger model instead—in this case, the full-size 175B parameter GPT-3 model (henceforth just "GPT-3"). That is, we use GPT-3 to self-amplify its own zero-shot translations to produce an initial dataset for distillation.

We now proceed to apply the same method as in Section 5.1 to all model sizes, but this time using few-shot samples from GPT-3 for the bootstrap. We display the evaluation BLEU scores throughout the bootstrap and after iterative backtranslation in Table 2. Interestingly, the higher-quality samples from GPT-3 appear to saturate the smaller models and they improve very little. Motivated by the possibility that our models are beginning to overfit to the WMT14 English-French training data, we attempt another experiment where 50% of the monotext for backtranslation is sampled from the English and French components of the CC100 dataset (Conneau et al., 2020). The extra monolingual data significantly benefits all model scales, improving English-French BLEU by approximately 3 points compared to iterative backtranslation on WMT data alone. With this setup, the `xl` attains a new unsupervised state-of-art of 42.1 BLEU on the WMT14 English-French benchmark.

## 6 DISCUSSION AND FURTHER ABLATIONS

**Bias towards English generation**   Previous work (Brown et al., 2020) has shown that after generative pre-training on a corpus of English-dominated Internet text, GPT-3 models are far more capable of translating into English than translating out of English. This is reflected by the disparity between English-French and French-English BLEU scores immediately after few-shot distillation and before backtranslation on the few-shot prompted data. Interestingly, after only two epochs of backtranslation on the relatively scarce few-shot prompted data, this gap is reversed, with all models achieving significantly higher English-French BLEU than French-English BLEU. The data efficiency of the bootstrap suggests that coming out of pre-training, the models are merely misaligned rather than deficient in knowledge about French, and that their latent knowledge about translation out of English can be surfaced using backtranslation. Relatedly, high-quality samples in one language in the previous round of backtranslation lead to higher-quality synthetic bitext for training the reverse direction in the next. This turns the asymmetry towards English generation into an advantage during backtranslation. However, if the initial disparity between the quality of the translation directions is extreme (as with the self-distilled `small`, which achieves $< 2$ BLEU for English-French few-shot compared to $\approx 10$ BLEU for French-English), then we see that the evaluation BLEU for either direction is unstable and oscillates between iterations, though they eventually converge upwards as backtranslation continues.

**Comparison to previous work**   In Table 3, we compare the BLEU scores attained by our best model (an `xl` distilled on self-amplified GPT-3 followed by 40 rounds of backtranslation) to prior

work in unsupervised neural machine translation on the WMT14 English-French benchmark. To ensure comparability to prior work, we report tokenized BLEU using `multi-bleu.perl` and the XLM tokenizer. This was used to report the few- and zero-shot performance of GPT-3 in Brown et al. (2020), which we also include in Table 3 for completeness.

|        | XLM  | MASS | CUNMT | XLM+ | CBD  | xl   | GPT-3 (fs) | GPT-3 (zs) |
|--------|------|------|-------|------|------|------|------------|------------|
| en-fr  | 33.4 | 37.5 | 37.6  | 40.2 | 38.2 | **41.7** | 32.6   | 25.2       |
| fr-en  | 33.3 | 34.9 | 35.2  | 36.9 | 35.5 | **38.0** | 39.2 | 21.2  |

Table 3: Comparison of our best model—an `xl` distilled on self-amplified GPT-3 followed by 40 rounds of iterative backtranslation—to prior work (Conneau & Lample, 2019; Song et al., 2019; Wang et al., 2021a; Keung et al., 2020; Nguyen et al., 2021) in unsupervised NMT on the WMT14 English-French benchmark. Bold indicates unsupervised state-of-the-art and underline indicates few-shot state-of-the-art.

## 6.1 ABLATING TEMPERATURE FOR FEW-SHOT DISTILLATION

|              |       | self-distill | backtrans. $\tau = 0.0$ | backtrans. $\tau = 0.3$ | backtrans. $\tau = 1.0$ |
|--------------|-------|--------------|-------------------------|-------------------------|-------------------------|
| $\tau = 0.0$ | en-fr | 20.3 | 34.4 | 34.7 | 27.8 |
|              | fr-en | 29.9 | 29.3 | 29.6 | 24.7 |
| $\tau = 0.3$ | en-fr | 20.6 | 33.9 | 35.1 | 27.6 |
|              | fr-en | 29.2 | 28.9 | 29.9 | 24.4 |
| $\tau = 1.0$ | en-fr | 20.2 | 34.9 | 34.6 | 27.6 |
|              | fr-en | 29.0 | 29.2 | 29.2 | 24.9 |

Table 4: English-French (top) and French-English (bottom) test BLEU using few-shot prompted samples generated with temperatures $\tau = 0.0, 0.3, 1.0$ throughout the bootstrap. We see that the temperature used for sampling has little effect on evaluation BLEU after few-shot distillation, while high-temperature samples are harmful during the backtranslation part of the bootstrap.

It was shown by Edunov et al. (2018) that backtranslation is more effective when the translations are slightly noisy, *i.e.*, sampled with nonzero temperature or via a noised beam search. This motivated our use of the temperature $\tau = 0.3$ throughout. We ablate this choice of temperature when sampling data for few-shot distillation, and study the effect of using $\tau = 0.0$ and $\tau = 1.0$ during the bootstrap using a `large` model. We display the results in Table 4. We see that lower temperatures lead to marginally higher test BLEU scores during distillation while $\tau = 1.0$ results in lower test *loss* and no overfitting after two epochs of training. However, regardless of the temperature of samples used for self-distillation, the differences in both test BLEU and test loss almost vanish after the backtranslation part of the bootstrap when training to backtranslate low temperature samples ($\tau = 0.0$ or $\tau = 0.3$).

## 6.2 FEW-SHOT SELF-AMPLIFICATION

We observed that few-shot prompting GPT-3 with its own zero-shot translations produced better translations than zero-shot prompting alone. We investigate this further by comparing the BLEU scores of zero-shot translations (sampled using the same prompt described in Section 4) to the BLEU scores of self-amplified few-shot prompted translations (*i.e.*, where the few-shot demonstrations are the zero-shot translations sampled from the same model) for all the model sizes studied in this paper. Our results are displayed in Table 5. We see that self-amplification improves translation quality at all model scales.

## 6.3 USING REAL FEW-SHOT EXAMPLES

So far our results have been completely unsupervised, but few-shot learning is typically studied in the context of semi-supervised learning (Wang et al., 2020), where the few-shot demonstrations

|  |  | small | medium | large | xl | GPT-3 |
|---|---|---|---|---|---|---|
| zero-shot | en-fr | 0.57 | 1.23 | 1.90 | 2.84 | 26.19 |
|  | fr-en | 2.00 | 13.92 | 8.14 | 19.60 | 25.49 |
| self-amplified | en-fr | 1.39 | 8.98 | 12.46 | 14.32 | 29.96 |
|  | fr-en | 5.76 | 16.75 | 21.75 | 23.98 | 31.75 |

Table 5: Zero-shot versus few-shot self-amplified test BLEU for all model sizes studied in this paper. For zero-shot generation we use the same prompt format described in Section 4. For self-amplified generation, we use the model's own zero-shot generations as in-context few-shot examples.

|  |  | small | medium | large | xl |
|---|---|---|---|---|---|
| few-shot ($\tau = 0.0$) | en-fr | 1.09 | 7.19 | 11.8 | 13.35 |
|  | fr-en | 3.86 | 14.58 | 20.34 | 23.01 |
| few-shot ($\tau = 0.3$) | en-fr | 1.09 | 6.83 | 11.38 | 13.08 |
|  | fr-en | 4.13 | 14.86 | 19.92 | 22.04 |
| few-shot ($\tau = 1.0$) | en-fr | 0.33 | 1.74 | 2.34 | 2.94 |
|  | fr-en | 0.94 | 4.18 | 4.64 | 7.25 |
| distillation | en-fr | 0.39 | 7.63 | 17.27 | 19.81 |
|  | fr-en | 3.9 | 20.29 | 27.65 | 30.89 |
| initial backtranslation | en-fr | 7.77 | 24.71 | 29.64 | 33.78 |
|  | fr-en | 1.7 | 18.9 | 26.61 | 30.93 |
| after backtranslation | en-fr | 31.23 | 34.42 | 37.86 | 39.39 |
|  | fr-en | 27.45 | 29.96 | 34.23 | 34.97 |

Table 6: English-French (top) and French-English (bottom) test BLEU throughout the few-shot self-distillation bootstrap across multiple model scales, this time using real few-shot examples. We see that performance after backtranslation is equivalent to that reported in Table 1.

|  |  | small | large |
|---|---|---|---|
| distillation | en-fr | 32.95 | 36.0 |
|  | fr-en | 32.45 | 36.29 |
| initial backtranslation | en-fr | 36.32 | 38.72 |
|  | fr-en | 32.43 | 36.61 |
| after backtranslation | en-fr | 36.38 | 39.36 |
|  | fr-en | 32.66 | 35.67 |
| after backtranslation (+CC100) | en-fr | 39.01 | 42.03 |
|  | fr-en | 34.17 | 36.94 |

Table 7: English-French (top) and French-English (bottom) test BLEU of the small and large models throughout the bootstrap and after iterative backtranslation, where for the bootstrap we use generations from 175B GPT-3 prompted using real few-shot examples. Similarly to Table 2, we observe a boost in final BLEU score when, after the bootstrap, we additionally sample monolingual text from the English and French portions of the CC100 dataset.

are real training data. In this section, we ablate the usage of synthetic few-shot translations in our methodology and reproduce our experiments from Section 5 using real few-shot demonstrations. We observe virtually no difference in BLEU score after iterative backtranslation.

We modify the few-shot prompting described in Section 5 as follows. Rather than sampling zero-shot translations for each half of our held-out pool of $N$=2048 training examples, we sample from these examples directly during few-shot prompting.

Table 6 displays test BLEU throughout the bootstrap and after iterative backtranslation for the same model sizes studied in Section 5.1. We see that our models converge to the same test BLEU (c.f. Section 5.1). Table 7 displays analogous results when distilling samples from GPT-3 with the `small` and `large` models, this time few-shot prompted using real examples. We again see that using real rather than synthetic few-shot demonstrations to sample the initial bootstrap data from GPT-3 has no effect on final BLEU score after iterative backtranslation.

### 6.3.1 ALMOST-UNSUPERVISED MACHINE TRANSLATION WITH THREE EXAMPLES ONLY

|       | $N$=3 | $N$=8 | $N$=16 | $N$=32 | $N$=64 | $N$=128 | $N$=256 | $N$=512 | $N$=1024 | $N$=2048 |
|-------|-------|-------|--------|--------|--------|---------|---------|---------|----------|----------|
| en-fr | 12.6  | 12.4  | 12.7   | 13.1   | 13.2   | 13.0    | 12.7    | 12.9    | 12.7     | 12.8     |
| fr-en | 21.5  | 21.3  | 22.1   | 22.4   | 21.9   | 22.3    | 22.1    | 22.1    | 22.2     | 22.1     |

Table 8: BLEU scores (calculated over 4096 random training examples) for the few-shot prompted translations from a `large` model, as the total number of available few-shot examples varies from $N = 3$ to $N = 2048$. We see that $N$ has minimal impact on the BLEU score of the sampled translations. Moreover, the difference in BLEU between the models bootstrapped using $N = 3$ versus $N = 2048$ disappears after iterative backtranslation.

Finally, we show that even in the semi-supervised setting, we can minimize the supervision available from few-shot demonstrations with no difference in test BLEU after backtranslation coverges. Table 8 displays the BLEU scores of few-shot sampled translations across various orders of magnitude of N, the number of available few-shot examples. Remarkably, even when N is decreased to 3, there is only a slight negative impact on the BLEU score of the few-shot sampled translations. We do not ablate lower values of $N$ in order to maintain the assumption of $k$=3 distinct in-context examples for few-shot prompting. We then run our entire procedure with a `large` model, using $N$=3 real few-shot demonstrations for the bootstrap followed by iterative backtranslation. We observe a final English-French BLEU of $38.0$ and French-English BLEU of $34.2$, on par with the final BLEU scores reported in Table 6.

## 7 CONCLUSION AND FUTURE DIRECTIONS

We remark that backtranslation, like reinforcement learning, is simply a way of exchanging compute for data. Instead of grounding the model with a reward signal from an environment, however, backtranslation exploits the symmetry of the translation task to ground the model by training it to cross-lingually denoise its own samples. Our present work can be viewed as part of a recent trend towards *data-driven architecture engineering*, where task-specific inductive biases, if any, are engineered into and learned from the training data instead of being hardcoded into the model architecture. In formulating the translation task in terms of language modeling, we see that the input-output inductive bias imposed by an encoder-decoder architecture can be simulated with prompt formatting. Similarly, we see that generative language modeling at sufficient scale combined with clever prompting for automated data generation can attain state-of-the-art results in unsupervised translation, rendering methods intended to produce strong encoders and aligned multilingual representations unnecessary.

Although we have focused solely on the domain of machine translation in this work, our methodology is applicable to any sequence-to-sequence task whose forwards and inverse directions are (1) to be jointly learned by an autoregressive decoder-only transformer and (2) are amenable to few-shot prompting after large-scale generative pre-training. Backtranslation is simply reverse self-training (Bojar & Tamchyna, 2011) and is fundamentally untied to the translation domain; we invite the research community at large to further explore this technique, moving beyond translation and towards applications reflecting the full generality of the transformer architecture.

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
