# OpenReview forum: "Unsupervised Neural Machine Translation with Generative Language Models Only"
_ICLR.cc/2022/Conference — ICLR 2022 Submitted_

### Official Review · Reviewer_LmFj · 2021-10-29

**Correctness:** 3
**Technical Novelty And Significance:** 1
**Empirical Novelty And Significance:** Not applicable
**Recommendation:** 3
**Confidence:** 5

**Main Review:**

Strengths:
 - State-of-the-art unsupervised translation results for both English-French and French-English.

Weakness:
 - Lack of novelty
 - Only one non-English language is considered.
 - No low-resource language pairs considered, which is the realistic setting for unsupervised MT.
 - No in-depth discussion on the aspects which separate this work from other unsupervised MT works.

The authors claim that "with sufficient scale, pre-training, and clever prompting, a single generative language model can implement the entire unsupervised neural machine translation pipeline, avoiding optimizations such as denoising autoencoding, auxiliary / adversarial losses in latent space, or ad-hoc bilingual dictionaries”. While I can agree that removing auxiliary / adversarial losses or ad-hoc bilingual dictionaries are certainly steps in the right direction, current state-of-the-art unsupervised translation models have already taken these steps and do not rely on these tools. For example, MASS [6] pre-trains using only a denoising autoencoding objective and yet achieves strong results comparable to the ones found in this paper (accounting for model size) and additionally does not rely on clever prompting. To be explicit, the MASS scores in Table 3 are comparable to the results of GPT-3 medium in Table 1, which brings into question if the gains found in this work are truly from the approach or simply from scaling.

It thus appears that the main contributions of this work which separate it from MASS are 1) using a decoder-only model rather than the typical encoder-decoder architecture, 2) using language modeling as pre-training objective instead of a denoising autoencoding objective and 3) using an English-centric model rather than a model appropriately trained on both languages. Of these, 1) is not particularly novel since GPT-3 has already demonstrated some initial translation performance for the language pairs in question. Previous work [1] has also shown that the initial quality of the translation system for iterative back-translation has little effect on the final model's quality. These two findings make it unsurprising that the authors achieved strong unsupervised translation performance after back-translation, despite the relatively weak model that was used to generate the few-shot examples or to initialize the back-translation process.  Moreover, other contemporary work [7] has already shown that decoder-only models are sufficient to build strong translation models and provided a far more thorough analysis of this architectural bias than this work.

For 2), it seems obvious that language modeling should result in a better pre-training objective, since the pre-trained model is directly trained to generate natural text. Models trained with masking strategies tend to struggle with zero-shot generation which hinders back-translation. For example, mBART [4] relied on LangID filtering in the initial stages of their back-translation routine to overcome this issue. The authors additionally claim that "[d]enoising autoencoding with a bilingual encoder can be viewed as a kind of latent bilingual lexicon induction, necessary for producing sufficiently aligned embeddings to kick-start back-translation." However, I see no reason why we can't view language modeling on multilingual data in the same light. The initial translation performance from GPT-3 already demonstrates that language model can serve as an acceptable substitute to denoising autoencoding for kick-starting back-translation.  With these thoughts in mind, we cannot consider this work’s successful removal of denoising autoencoding for training unsupervised translation models as a particularly difficult task nor a surprising finding but instead as more of a confirmation of an expected result.

The English-centric nature of the model (point 3) is an interesting aspect, however it is understudied in the paper. The choice of language pair also hinders this study. For example, French is the second most populous language in GPT-3’s data [8]. Moreover, the original authors of GPT-3 have already demonstrated that GPT-3 can perform English < - - > French translation,  further reducing the impressiveness of the result. It is also interesting to note that this English bias the authors comment on also affects traditional multilingual machine translation models (both supervised [2] and unsupervised [3]) due to the reliance on English-centric parallel data. However, in this setting, this bias is even more pronounced since the English data does not arise from translation data. It would be interesting to explore whether the overabundance of English monolingual data (or perhaps simply scale?) is enough to make up for the scarcity of data available for some of the low-resource languages which appear in GPT-3’s corpus. Similarly, it could be interesting to see if the scripts of the languages considered or their similarity with English is an important feature for this approach to succeed. For example, do we expect GPT-3 to also excel at Hindi < - - > English translation? Or Gujarati < - - > English? Both languages appear in GPT-3’s pre-training corpus, so one could test it. The performance on these languages is crucial because the true value of unsupervised translation lies in low-resource language pairs, and their omission puts into question the validity of this method in realistic settings. The absence of true low-resource language pairs for evaluating unsupervised translation is unacceptable, especially when considering most of the recent literature in unsupervised translation includes such language pairs.

Lastly, it would also be interesting to explore whether these models are able to generate more natural text, avoiding the pitfalls of translationese that traditionally affects models trained with supervised data.

Questions for authors:

1. What is [[TRANSLATE]] in Figure 1? Is this just a stand-in for the prompt written in 4.1?

2. Many approaches in multilingual MT tend to discard the source language and use only the target language when biasing the model for translation. Yet it seems like you used the source language for both prompting GPT-3 and for the zero-shot format in the distillation. Leveraging the source language name could inhibit translation of code-mixed text. Did you experiment using only the target language instead in the prompt or attempted any investigation on whether including the source language name affected the translation performance?

3. Did you attempt this for non-English-centric language pairs?

Citations:

[1] Artetxe et. al. Do all Roads Lead to Rome? Understanding the Role of Initialization in Iterative Back-Translation
[2] Arivazhagan et. al. Massively Multilingual Neural Machine Translation in the Wild: Findings and Challenges
[3] Garcia et. al. Harnessing Multilinguality in Unsupervised Machine Translation for Rare Languages
[4] Liu et. al. Multilingual Denoising Pre-training for Neural Machine Translation
[5] Siddhant et. al. Leveraging Monolingual Data with Self-Supervision for Multilingual Neural Machine Translation
[6] Song et. al. MASS: Masked Sequence to Sequence Pre-training for Language Generation
[7] Wang et. al. Language Models are Good Translators
[8] https://github.com/openai/gpt-3/blob/master/dataset_statistics/languages_by_word_count.csv

**Summary Of The Paper:**

This work studies the use of a large English-centric language model (GPT-3) for the purposes of building an unsupervised machine translation system for the language pairs English-French and French-English. Given some monolingual data, the authors use GPT-3 to generate zero-shot translations, which then get used as few-shot examples for a smaller generative model to generate a synthetic translation dataset. This synthetic dataset is used to fine-tune the language model for translation through a back-translation scheme. With this approach, the authors are able to attain a new state-of-the-art result for unsupervised translation on the language pair English-French.

**Summary Of The Review:**

In summary, I don’t believe this work in its current shape is of the adequate quality to be published at ICLR. The biggest flaw is that it is unclear whether anything presented in the work is truly novel or unknown to the machine translation community. The work can be jointly summarized as “language modeling can yield an initial translation model whose performance can boosted tremendously by back-translation”, but no single aspect of this is new, nor is the success of the combination of the various pieces at all surprising to anyone who has been following the unsupervised translation literature over the past two years.

Coupled with the limitations of only studying a single non-English language, no low-resource language pairs, as well no detailed discussion/experiments on the few aspects which distinguish it from other works (e.g. the overwhelming English-centric bias of GPT-3), I cannot recommend acceptance.

---

> ### Author Response · Authors · 2021-11-23
> **Response to Reviewer LmFj**
>
> Thank you for your review. We first address the questions raised at the end of the review.
>
> > What is [[TRANSLATE]] in Figure 1? Is this just a stand-in for the prompt written in 4.1?
>
> It is literally part of the prompt shown to the language model. We never train on the prompt written in section 4.1; it is only used to elicit zero-shot translations from GPT-3. During backtranslation, we train on and sample translations using the prompt displayed in Figure 1 and described in Section 3: bitext ⟨seq1, seq2⟩ is formatted as `[L1] <seq1> [[TRANSLATE]] [L2] <seq2>`.
>
> > Many approaches in multilingual MT tend to discard the source language and use only the target language when biasing the model for translation. Yet it seems like you used the source language for both prompting GPT-3 and for the zero-shot format in the distillation. Leveraging the source language name could inhibit translation of code-mixed text. Did you experiment using only the target language instead in the prompt or attempted any investigation on whether including the source language name affected the translation performance?
>
> Thank you for suggesting this ablation. First, we would like to clarify that the zero-shot prompt described in Section 4.1 is only used to elicit the initial zero-shot translations from GPT-3. Throughout backtranslation, we use the prompt formatting `[L1] <seq1> [[TRANSLATE]] [L2] <seq2>`. So, the question is really about how the language used for a zero-shot prompt affects the translation ability of just a pre-trained GPT-3 model.
>
> To investigate this question, we evaluated the zero-shot translation ability of a GPT-3 `large` model using the following prompt in French:
>
> ```
> Étant donné le passage suivant dans {src_lang} :{separator}{src_seq}{separator}une bonne traduction {tgt_lang} est :{separator}
> ```
> where `src_lang`, `tgt_lang` are either "anglais" or "français".
>
> We observed en->fr and fr->en BLEU scores of 5.16 and 4.40, compared to 1.90 and 8.14 zero-shot BLEU scores obtained with the English prompt reported in Table 5 of the paper, consistent with the hypothesis that prompts written with the target-side language improve source-to-target translation. We are happy to expand this analysis to more model sizes and measure its effect during the initial rounds of backtranslation with smaller models.
>
> > Did you attempt this for non-English-centric language pairs?
>
> We did verify using smaller models that our methodology works as expected on English-Romanian and English-German, but did not attempt to push state-of-the-art. Due to compute constraints, we were unable to run all variants of our experiments on other language pairs, and omitted them to clarify the presentation.
>
> We now address some other points from earlier in the review.
>
> > While I can agree that removing auxiliary / adversarial losses or ad-hoc bilingual dictionaries are certainly steps in the right direction, current state-of-the-art unsupervised translation models have already taken these steps and do not rely on these tools.
>
> > It thus appears that the main contributions of this work which separate it from MASS are 1) using a decoder-only model rather than the typical encoder-decoder architecture, 2) using language modeling as pre-training objective instead of a denoising autoencoding objective and 3) using an English-centric model rather than a model appropriately trained on both languages.
>
> > To be explicit, the MASS scores in Table 3 are comparable to the results of GPT-3 medium in Table 1, which brings into question if the gains found in this work are truly from the approach or simply from scaling.
>
> Thank you for raising these points. We agree that in our work it is presently unclear whether using language modeling as a pre-training objective is more efficient than other pre-training objectives for bootstrapping an unsupervised NMT system, and that translation with other non-English languages could be better explored. We emphasize that the main contribution of our work is showing how to use only autoregressive language modeling (via pre-training, zero-shot prompting, distillation, and backtranslation) to produce state-of-the-art unsupervised NMT systems. There is no prior work that accomplishes this.
>
> We would also like to emphasize that regardless of whether the gains in this work are from the approach or simply from scaling, an approach like ours which uses only autoregressive language modeling is desirable because it is simpler. As pointed out by the MASS authors, the autoregressive language modeling objective can be thought of as a special case of the MASS objective where the masking is trivial. One could interpret our results as saying that, controlling for model size, this simpler approach can attain equivalent performance as the more complicated MASS objective, and can furthermore be scaled to achieve state-of-the-art.

---

### Official Review · Reviewer_K8G8 · 2021-11-02

**Correctness:** 3
**Technical Novelty And Significance:** 3
**Empirical Novelty And Significance:** Not applicable
**Recommendation:** 6
**Confidence:** 4

**Main Review:**

Strengths:
1. A new approach to construct UNMT with generative pre-trained language models.
2. The proposed method achieves the new start-of-the-art unsupervised translation performance on the WMT14 English-French benchmark.

Weaknesses:
1. This paper only verifies the effectiveness of the proposed method on the English-French language pair. It is better to conduct experiments on other language pairs, such as English-German, to make the result more reliable.
2. Currently, the performance of UNMT model on distant language pair is very poor. It would be more exciting, if we also observe the significant improvement of the proposed method on some distant language pairs, such as English-Chinese and English-Japanese.

Detailed Comments:

Overall, this paper is easy to follow and understand. I like this idea of constructing UNMT system with the GPT-X model and the proposed method is very interesting. Also, the experimental results on WMT14 English-French benchmark surprise me.

My main concern is the experiment settings. This paper merely verifies the effectiveness of the proposed method on the English-French language pair, making the experiment results not convincing enough. It is better to conduct experiments on other language pairs, such as English-German. Previously, the performance of UNMT methods on distant language pair is very poor. As the proposed method replaces the bilingual dictionary with few-shot amplification, it seems to me that the proposed method may achieve better performance on distant language pairs. I am looking forward to observing the unsupervised translation performance of the proposed method on some distant language pairs.

Questions for the Author(s):
1. As the proposed method requires large-scale model to achieve the start-of-the-art performance, what is the training cost/time?
2. In fact, this proposed method could be applied in sequence-to-sequence pretraining model, such as mT5. Have you explored this direction? Or leveraging synthetic parallel data generated by the GPT-3 to warm up the previous UNMT models?

Missing Reference:
1. Unsupervised Neural Machine Translation with Weight Sharing. Yang et al., ACL 2018
2. Phrase-Based & Neural Unsupervised Machine Translation. Lample et al., EMNLP 2018
3. Unsupervised Neural Machine Translation with SMT as Posterior Regularization. Ren et al., AAAI 2019
4. Extract and Edit: An Alternative to Back-Translation for Unsupervised Neural Machine Translation. Wu et al., NAACL 2019
5. An Effective Approach to Unsupervised Machine Translation. Artetxe et al.,  ACL 2019



**Summary Of The Paper:**

This paper investigates the way of constructing the UNMT model with a generative pre-trained language model, such as the GPT-X model. This method consists of few-shot amplification, distillation and back-translation, in which the first two steps are leveraged to warm up a UNMT model instead of bilingual dictionaries inferred in an unsupervised way. When using GPT-3 to obtain synthetic parallel data for model training through the zero-shot prompting approach, the proposed method achieves the state-of-the-art unsupervised translation performance on the WMT14 English-French benchmark.

**Summary Of The Review:**

This paper is easy to follow, and the idea is very interesting and novel. The proposed method achieves the new start-of-the-art unsupervised translation performance on the WMT14 English-French benchmark. It is better to conduct additional experiments on other language pairs to make the experiment results more reliable.

---

> ### Author Response · Authors · 2021-11-23
> **Response to Reviewer K8G8**
>
> Thank you for your review.
>
> > As the proposed method requires large-scale model to achieve the start-of-the-art performance, what is the training cost/time?
>
> All models in the paper are trained for 32 billion tokens for the bootstrap and then for 80 billion tokens cumulatively after backtranslation. In comparison, the GPT-3 models were pretrained for 300 billion tokens, so training costs are approximately 25% of that reported for pre-training the GPT-3 models, e.g. up to 7 petaflop-s/days for our `xl` model. Exact numbers can be found in Table D.1 of the GPT-3 paper [1].
>
> > In fact, this proposed method could be applied in sequence-to-sequence pretraining model, such as mT5. Have you explored this direction? Or leveraging synthetic parallel data generated by the GPT-3 to warm up the previous UNMT models?
>
> In our present work we have focused explicitly on UNMT using only generative language modeling, and have not explored using synthetic data from GPT-3 to warm up other encoder-decoder UNMT models. We agree that this is a promising method for data augmentation and would like to see it explored further.
>
> > It is better to conduct experiments on other language pairs, such as English-German, to make the result more reliable.
>
> We did verify using smaller models that our methodology works as expected on English-Romanian and English-German, but did not attempt to push state-of-the-art. Due to compute constraints, we were unable to run all variants of our experiments on other language pairs, and omitted them to clarify the presentation.
>
> [1] https://arxiv.org/abs/2005.14165

---

### Official Review · Reviewer_bUnG · 2021-11-02

**Correctness:** 3
**Technical Novelty And Significance:** 3
**Empirical Novelty And Significance:** 3
**Recommendation:** 5
**Confidence:** 3

**Main Review:**

Interesting results, but I think this study is already standing at other position than so-called "unsupervised translation" because it uses pretrained LM as a seed and thus the model is based on a huge amount of additional resources. It would be not reasonable to say state-of-the-art by comparing its accuracy with other "unsupervised" models trained from only a specific corpus.
Table 8 is also interesting: in this task we can only prepare monolingual corpora for each language/domain and only a few seed translations for prompts (if we didn't generate them from LMs).

Since pretrained LMs are trained using a bunch of resources available from the Internet, its training data possibly includes the training/evaluation data used in this experiment, meaning that there would be some suspicion about data leakage. Is it checked appropriately that the training data were not leaked, or could you say the experiment is actually safe without this check for some reason?

Overall, the text is hard to read to me. Many terms were introduced without definitions, the sentence frequently omitted important contexts, and syntax was often broken.

**Summary Of The Paper:**

The paper proposed an iterative finetuning method to adapt pretrained LMs to translation tasks. The framework refines the LM into translation models, which accepts both languages with a predefined format, and translate them into other language bidirectionally.
The model is finetuned by backtranslation.

To generate translation counterparts of the given monolingual corpora during finetuning, the paper utilized prompting on LMs: gives several translations X1 Y1 X2 Y2 ... as contexts, then predict the successor Yn given the real input Xn, where Xk and Yk are souce/target sentences.

The context of the prompt X1 Y1 X2 Y2 is also generated by LM using another template, but this part can be replaced as a real examples.

Experiments show that the trained system achieves the better performance than existing similar methods, showing that the effectiveness of the proposed method. The paper provided some additional studies to support the main results.

**Summary Of The Review:**

- The paper employed huge LM/prompting/backtranslation techniques to construct bidirectional translation models. The technique of model construction seems remarkable, and there is some concern about data.
- The paper is somewhat hard to read due to syntactic/semantic issues, and I guess I possibly drops some context to assess the paper correctly.
- I am not fully familiar with recent advances in so-called "unsupervised translation" and huge LMs, and I didn't judge if the proposed method is novel or not, and not provided comparison with existing studies.

---

> ### Author Response · Authors · 2021-11-23
> **Response to Reviewer bUnG**
>
> Thank you for your review.
>
> > Interesting results, but I think this study is already standing at other position than so-called "unsupervised translation" because it uses pretrained LM as a seed and thus the model is based on a huge amount of additional resources. It would be not reasonable to say state-of-the-art by comparing its accuracy with other "unsupervised" models trained from only a specific corpus.
>
> Thank you for raising this point. It is not uncommon for other unsupervised models to leverage additional training data outside of the target benchmark. For example, recent XLM models [2] are pre-trained on CommonCrawl, the original XLM model (the basis for CBD and XLM+ reported in our Table 3) was pre-trained on multilingual Wikipedia, and MASS was trained on monolingual text from multiple WMT benchmarks.
>
> > Is it checked appropriately that the training data were not leaked, or could you say the experiment is actually safe without this check for some reason?
>
> WMT14 training data was not part of the training mix for GPT-3. A test-set contamination study including the WMT14 French-English benchmark was also reported in Appendix C of the GPT-3 paper [1]. It found that test BLEU on WMT14 English-French actually slightly *improved* when all potentially contaminated test examples were removed from the evaluation, suggesting negligible effect from train-test contamination.
>
> [1] https://arxiv.org/abs/2005.14165
> [2] https://arxiv.org/abs/1911.02116

---

### Official Review · Reviewer_LnSu · 2021-11-07

**Correctness:** 1
**Technical Novelty And Significance:** 2
**Empirical Novelty And Significance:** 2
**Recommendation:** 5
**Confidence:** 3

**Main Review:**

Strengths:
- This is an interesting way to use large pre-trained language models such as GPT-3 to solve generative tasks like Machine Translation
- Combines multiple recipes from various sources (e.g. zero shot MT, few shot MT, back-translation) and shows how to apply them to a standard left-to-right pre-trained LM to achieve impressive translation results

Weaknesses:
- While the results are interesting and illustrate the flexibility of a standard transformer language model, the key ingredient of this approach relies on having a language model that has seen a very large and diverse dataset that very likely includes some translation examples (for zero-shot translation to work). Using such a language model to boot-strap self-training (called distillation in this paper) followed by back-translation is interesting, but should be grounded in techniques that have access to the same kind of large and diverse dataset. One clear example is bi-text mining (e.g. see https://arxiv.org/abs/1911.04944) where one can simply mine a bunch of parallel text from a similar kind of dataset and then train a standard encoder-decoder machine translation model (with self-training / backtranslation if so desired). It is not clear what the advantage of the proposed method over such an approach would be except as a curious application of existing pre-trained models. Could the authors elaborate on this point and, if they agree, compare to the methods described above.
- Are there other works on decoder-only machine translation? (i.e. ones that do not have a decoder and rely on only the LM part but trained on bi-text data). If so, this would be great to discuss in related work.

**Summary Of The Paper:**

This paper proposes to solve the machine translation task without any bi-text data and using only a pre-trained generative language model to bootstrap the process. The entire method relies on using powerful pre-trained language models that have been trained on a large amount of data to extract initial boot-strap examples, and then fine-tune the model to solve the machine translation task using specially formatted prompts. The same language model is used to translate in both directions, and back-translation is used to improve results.

**Summary Of The Review:**

While this work is an interesting way to explore the power of large pre-trained language models, I am not sure it is compared properly to methods (also starting with no parallel data) that may have access to the same kind of datasets used to train these language models. I am also not sure how this work compares to supervised and/or semi-supervised techniques using the same model sizes, or what is the state of the art for using transformer LMs with prompting for NMT (where parallel data is available). Discussion of these points would be enlightening and may lead to updating my recommendation.

---

> ### Author Response · Authors · 2021-11-23
> **Response to Reviewer LnSu**
>
> Thank you for your review.
>
> > Using such a language model to boot-strap self-training (called distillation in this paper) followed by back-translation is interesting, but should be grounded in techniques that have access to the same kind of large and diverse dataset. One clear example is bi-text mining (e.g. see https://arxiv.org/abs/1911.04944) where one can simply mine a bunch of parallel text from a similar kind of dataset and then train a standard encoder-decoder machine translation model (with self-training / backtranslation if so desired). It is not clear what the advantage of the proposed method over such an approach would be except as a curious application of existing pre-trained models.
>
> Thank you for raising this point. One way to interpret our results is that we can view large-scale pre-training as implicit bi-text mining, and that synthetic bitext can be extracted from pre-trained language models using zero- and few-shot prompting. The main advantage of this approach is its simplicity. Synthetic (and potentially more diverse, as it is sampled rather than extracted) bi-text can be obtained from a pre-trained model without any additional engineering or having to optimize a separate bitext mining pipeline. Our method is also potentially more data-efficient. In comparison to the linked work, which extracts and trains on 94.1M English-French pairs, we use zero- and few-shot prompting to extract only 8M translations to bootstrap our models before running backtranslation.
>
> > I am also not sure how this work compares to supervised and/or semi-supervised techniques using the same model sizes, or what is the state of the art for using transformer LMs with prompting for NMT (where parallel data is available). Discussion of these points would be enlightening and may lead to updating my recommendation.
>
> There is contemporaneous work on *supervised* NMT with decoder-only transformers: https://arxiv.org/pdf/2106.13627.pdf. Their best model (LM4NMT-Big) has approximately 300M parameters, comparable to GPT-3 medium. It achieves an en->fr BLEU score of 42.9, slightly higher than our best unsupervised model.

---

### Decision · Program_Chairs · 2022-01-20

**Decision:**

Reject

**Comment:**

This paper proposes an alternative method to improve UNMT by using only a pre-trained generative language model to bootstrap the process. In all, the reviewers think the proposed method is reasonable.

However, the empirical part is not convincing.  Most reviewers think that evaluating the method on one language pair (En-Fr) is not enough to show the effect of the proposed method. In addition, some reviewers argue the clarity of this paper.

In all, I think the proposed method is meaningful. However, the current version is not ready to be published in this ICLR. I hope the authors can improve their paper according to the reviews.